# Consistency of Trans-Abdominal and Water-Immersion Ultrasound Images of Diseased Intestinal Segments in Crohn’s Disease

**DOI:** 10.3390/diagnostics10050267

**Published:** 2020-04-29

**Authors:** Feiqian Wang, Kazushi Numata, Hiromi Yonezawa, Kana Sato, Yoshito Ishii, Katsuki Yaguchi, Nao Kume, Yu Hashimoto, Masafumi Nishio, Yoshinori Nakamori, Aya Ikeda, Akira Madarame, Atsuhiro Hirayama, Tsuyoshi Ogashiwa, Tomohiko Sasaki, Misato Jin, Akiho Hanzawa, Naomi Shibata, Shinichi Hashimorto, Yusuke Saigusa, Yoshiaki Inayama, Shin Maeda, Hideaki Kimura, Reiko Kunisaki

**Affiliations:** 1Gastroenterological Center, Yokohama City University Medical Center, 4-57 Urafune-cho, Minami-ku, Yokohama 232-0024, Japan; wangfeiqian@126.com (F.W.); m.nishio@hotmail.co.jp (M.N.); 2Ultrasound Department, The First Affiliated Hospital of Xi’an Jiaotong University, No. 277 West Yanta Road, Xi’an 710061, China; 3Department of Laboratory Medicine and Clinical Investigation, Yokohama City University Medical Center, Yokohama 232-0024, Japan; yone_317@yahoo.co.jp (H.Y.); jin33@yokohama-cu.ac.jp (M.J.); ace_base603@yahoo.co.jp (A.H.); n_sibata@yokohama-cu.ac.jp (N.S.); 4Inflammatory Bowel Disease Center, Yokohama City University Medical Center, Yokohama 232-0024, Japan; v038eb@yamaguchi-u.ac.jp (K.S.); e123010g@yokohama-cu.ac.jp (Y.I.); e103084c@yokohama-cu.ac.jp (K.Y.); e143030g@yokohama-cu.ac.jp (N.K.); take_it_easy1996@yahoo.co.jp (Y.H.); osaehiro@yokohama-cu.ac.jp (Y.N.); ikedaa@yokohama-cu.ac.jp (A.I.); 12madarame21@gmail.com (A.M.); a_hira@yokohama-cu.ac.jp (A.H.); viva.kappa.byakui.oh-yeah@plum.plala.or.jp (T.O.); ssktmhk@yokohama-cu.ac.jp (T.S.); hkim@yokohama-cu.ac.jp (H.K.); 5Department of Gastroenterology and Hepatology, Yamaguchi University Graduate School of Medicine, Ube, Yamaguchi 755-8505, Japan; has-333@yamaguchi-u.ac.jp; 6Department of Gastroenterology, Yokohama City University Graduate School of Medicine, 3-9 Fukuura, Kanazawa-ku, Yokohama 236-0004, Japan; smaeda@yokohama-cu.ac.jp; 7Department of Biostatistics and Epidemiology, Graduate School of Medicine, Yokohama City University, Yokohama 236-0004, Japan; saigusay@yokohama-cu.ac.jp; 8Division of Diagnostic Pathology, Yokohama City University Medical Center, Yokohama 232-0024, Japan; inayama@yokohama-cu.ac.jp

**Keywords:** Crohn’s disease, water-immersion ultrasound, trans-abdominal ultrasound, surgical specimen

## Abstract

The aim of this study is to clarify whether trans-abdominal ultrasound (TAUS) can reflect actual intestinal conditions in Crohn’s disease (CD) as effectively as water-immersion ultrasound (WIUS) does. This retrospective study enrolled 29 CD patients with 113 intestinal lesions. Five ultrasound (US) parameters (distinct presence/indistinct presence/disappearance of wall stratification in the submucosal and mucosal layers; thickened submucosal layer; irregular mucosal surface; increased fat wrapping around the bowel wall; and fistula signs) that may indicate different states in CD were determined by TAUS and WIUS for the same lesion. Using WIUS as a reference standard, the sensitivity, specificity, and accuracy of TAUS were calculated. The degree of agreement between TAUS and WIUS was evaluated by the *kappa* coefficient. All US parameters of TAUS had an accuracy >70% (72.6–92.7%). The highest efficacy of TAUS was obtained for fistula signs (sensitivity, specificity, and accuracy values were 63.6%, 96.0%, and 92.7%, respectively). All US parameters between TAUS and WIUS had a definitive (*p* ≤ 0.001) and moderate-to-substantial consistency (*kappa* value = 0.446–0.615). The images of TAUS showed substantial similarity to those of WIUS, suggesting that TAUS may function as a substitute to evaluate the actual intestinal conditions of CD.

## 1. Introduction

Crohn’s disease (CD) is a chronic inflammatory bowel disease (IBD) characterized by ulceration and transmural inflammation, and its incidence is increasing worldwide [1]. This chronic, life-long, costly disease cannot be completely cured, and there is often a period of relapse or deterioration in the clinical course of CD. It is therefore crucial to apply a modality to accurately, cost-effectively, and repeatedly assess the progress of CD over life-long follow-up.

Endoscopic examination, the gold standard for CD, may cause patient discomfort. More importantly, CD is a transmural disease, but endoscopy only allows for visualization of the mucosa [2]. CT and magnetic resonance enterography have the disadvantages of the need to use a contrast agent, colonic luminal distension, or bowel cleaning, which requires a long period of preparation and causes discomfort to the patient [3]. Because patients suffering from CD are frequently diagnosed at a relatively young age and require repeated imaging, exposure to large cumulative doses of CT radiation at an early age might increase the risk of cancer [4]. Magnetic resonance imaging is not widely available in all clinical centers because it is time-consuming and expensive [5]. The newly released guideline of the European Crohn’s and Colitis Organization (ECCO) has placed emphasis on the role of trans-abdominal ultrasound (TAUS) as a radiation-free, non-invasive, and inexpensive imaging modality for the diagnosis of suspected CD because it can determine the activity and therapeutic response of transmural inflammation in CD [6]. TAUS is a potential alternative to endoscopy and offers the opportunity to detect and stage inflammatory, obstructive, and fistulizing CD [7]. More importantly, TAUS has a promising value in the detection of extramural complications, as well as the prediction of postoperative surgical recurrence in CD [8]. In many European countries, intestinal TAUS is becoming the first-line imaging method for patients with suspected inflammatory bowel disease, especially CD [9]. Many patterns, such as increased wall thickness, loss of wall stratification, small bowel dilation, mesenteric enlargement, and fistula signs, are considered to have diagnostic significance for CD to some extent [5].

Many studies have used TAUS to examine ultrasound (US) features to explore the relationship between US features and histological changes in the diseased intestine of CD patients. However, heterogeneous results using TAUS related to undetermined factors, such as examination position, differences between operators, and different diagnostic cutoff values, have frequently been reported [10,11,12,13]. Moreover, images of TAUS are considered to have a low quality under unfavorable conditions, such as gas obstruction, intestinal peristalsis, and abdominal wall fat, especially in patients with a large body size or when the intestinal lesion is located deep within tissues [2]. Therefore, whether TAUS can truly be used to detect the US parameters of CD and, further, to reflect the relationship between US parameters and histopathological changes needs to be determined.

Water-immersion ultrasound (WIUS) can be used directly on intestinal specimens obtained by surgical resection, and it is thought to provide high-quality images. Kunihiro et al. used WIUS to successfully identify wall stratification in 91.3% of CD lesions [14]. Very recently, Yaguchi et al. also found that WIUS clearly visualized the thickness and hierarchical structure of a bowel segment involving CD [15]. Although WIUS is highly sensitive for diagnosis, the inconvenience of its in vitro study limits its clinical application compared with TAUS.

The current state of US diagnosis for CD, especially TAUS, suggests the need to explore whether, and to what degree, TAUS reflects the appearance of images obtained by WIUS, and whether it can replace WIUS. Encouragingly, Hata et al. compared wall stratification and thickness obtained by WIUS and TAUS in 22 bowel segments, including those of normal bowel and IBD, and reported that the two methods closely corresponded with each other for different types of IBD [16]. Unfortunately, few US parameters were used, and the sample size was small; therefore, the conclusion of this preliminary study needs to be verified.

Taken together, using WIUS as a reference standard, we determined whether TAUS is equivalent to WIUS for diagnosis. This analysis may help assess whether images obtained by TAUS correspond to histological alterations in CD.

## 2. Materials and Methods

### 2.1. Patients

Overall, 113 consecutive intestinal surgery specimens from 29 patients were included in this study. This retrospective study was approved by the Institutional Review Committee and complied with the principles of the Helsinki Declaration. Informed consent was obtained from each participating patient (patient consent statement no. B190200027).

The main inclusion criteria for the present study were as follows:

A reliable or definite diagnosis of CD could be obtained, and the diseased intestinal segment was resected in surgery. According to the criteria of the Japanese Research Committee on Inflammatory Bowel Disease [15], a diagnosis of CD was made by gastroenterologists on the basis of history, physical exam, laboratory/radiological studies, and gastrointestinal histology. From December 2006 to December 2017, enterectomy was performed in 396 patients with a definite diagnosis of CD in our hospital.

TAUS was performed within 2 weeks before enterectomy. Of the 396 patients, 123 underwent TAUS within the 2 weeks prior to their operation.

Most patients did not undergo TAUS within two weeks. One reason was that the diagnosis obtained from TAUS two weeks previously already showed the necessity for a resection operation (such as serious stenosis, refractory, and fistula); therefore, there was no need for a re-examination by TAUS before surgery. Another reason was that emergency intestinal resection was performed in some patients because of unexpected intestinal perforation or massive accidental bleeding. A routine preoperative TAUS examination would cause a delay at that critical moment.

WIUS was performed within 30 min after enterectomy. Twenty-nine patients met these conditions (Table 1).

### 2.2. Preoperative TAUS Examinations

The patient was examined in the supine position to help relax the abdominal muscles and reduce the antero-posterior diameter of the abdomen. The scan started with a longitudinal view of the entire intestine, providing a broad overview to assess the extent of the affected bowel. The terminal ileum was clearly identified in all patients by confirming its connection with the cecum, and the ileum was identified as bowel loops in the middle-right lower quadrant. When any abnormality was found in the longitudinal section, the examiners used a cross-section scan to make a specific observation. Aplio XG, XV and Aplio 500 US systems (Canon Medical Systems, Otawara, Japan) equipped with native tissue harmonic grayscale imaging were used. Generally, a convex probe with a frequency of 3.5–6.0 MHz was used for TAUS examination. A more localized and detailed scan of the affected area was performed using a 7.5–12.0 MHz linear probe. To avoid operator discrepancy, TAUS was conducted and lesions were determined by two experienced sonographers (NS and AH) who had more than 5 years of experience in bowel US. For surgery convenience and accuracy, the examiners recorded the location of lesions in the TAUS examination report. TAUS images and videos were stored under the patient’s ID record. The examiners were not blinded to the clinical parameters of the patient.

### 2.3. Resection of Intestinal Segments

Patients who underwent intestinal surgery fasted 1 day before resection. At laparotomy, with reference to the results of preoperative endoscopy and imaging examinations, surgeons sequentially palpated all segments of the patient’s intestine to identify serious stiffness and stricture segments that should be removed. The excision was performed by a highly experienced surgeon who had been working for 20 years in the IBD field (HK). The resected specimens were incised longitudinally along the antimesenteric border, and the observed macroscopic findings were described in the surgical report. To better determine the anatomical location of the target lesion, the doctor performing WIUS in the next step was present during surgery and photographed the appearance of the intestinal segment before and after resection under the condition of laparotomy.

### 2.4. WIUS Examination

To prevent denaturation of the resected tissue, WIUS examinations were performed on diseased bowel segments within 30 min after surgical resection. Resected segments were pinned in a specimen container filled with cold physiological saline solution before being examined by WIUS.

To ensure that the lesions observed in vivo were the same as those that were surgically resected, we took the following measures in the WIUS examination. For lesions in and around the ileocecum, we measured the distance between the target lesion and the ileocecal valve. Lesions located in the colon, which is far from the ileocecum, were identified according to a comparison of preoperative TAUS findings and surgical photos. If it was not certain whether the surgically resected segment was in the same location as that targeted by US, the sample was excluded from WIUS examination and subsequent analysis. The segments analyzed by WIUS in this study were not necessarily the primary lesions that necessitated surgery. For example, even if surgery was performed for an internal fistula, we included the macroscopically intact areas and/or longitudinal ulcer scars in the resected intestinal specimens.

US scans of each segment were performed using an Aplio XG and Aplio 500 US system with a 5.0–11.0 MHz high-frequency linear probe. The US parameters were set as follows: gain: 85–95; depth: 4–6 cm; dynamic range: 65 dB; and focus position: around the region of interest. Differential tissue harmonic imaging and spatial compound images were used to obtain sufficient quality for evaluation. To avoid operator dependence, WIUS examinations were performed in cooperation with a sonographer with 5 years of experience in IBD ultrasound examination (TS) and a gastroenterologist (RK) who specialized in IBD. The intestinal tract, which was incised and opened on the anti-mesenteric border, was carefully scanned in open view (observed from the mucosal surface) and closed view, from longitudinal and transverse views of the bowel section, respectively. WIUS images and videos were stored under the patient’s ID record.

### 2.5. Baseline Data Processing and Ultrasonic Feature Extraction

Before evaluating the usefulness of TAUS, the two doctors who examined WIUS in the previous step (TS, RK) matched images of the same lesion by carefully reviewing the images and the video of TAUS and WIUS and photos of the appearance of the intestinal segment before and after laparotomy. Then, they identified and recorded the US changes from the images and videos of TAUS and WIUS according to previously published US parameters [14,16,17,18] and our clinical experience in CD diagnosis, including the following:

(1) Distinct presence (Figure 1, Figure 2d,e on TAUS)/indistinct presence (Figure 3, Figure 2b,c on WIUS)/disappearance (Figure 4 and Figure 5) of wall stratification in the submucosal and mucosal layer. As we show in Figure 6, the US image allows for the visualization of the five layers of the bowel: the border between the lumen and mucous layer (hyperechogenic), the mucous layer (hypoechogenic), the submucosa (hyperechogenic), the muscular layer (or the muscle membrane proper) (hypoechogenic), and the serosa layer (hyperechogenic) [19]. We used stratification-related US parameters to observe the border line between the mucosal and submucosal layers because discriminating between these two layers is meaningful for the evaluation of CD. We did not judge the border line between the submucosal and muscular layers.

(2) Thickened submucosal layer (Figure 1 and Figure 3). Thickened submucosal layer was defined as submucosal thickness accounting for more than 40% of the total intestinal wall. This US parameter was evaluated using lesions in the “presence of wall stratification in the submucosal and mucosal layers” in the first parameter, but not lesions with a “disappearance of wall stratification in the submucosal and mucosal layers”.

(3) Irregularity of mucosal surface (Figure 2, Figure 3, Figure 4 and Figure 5).

(4) Increase in fat wrapping (Figure 2, Figure 3 and Figure 4). Bright white hyperechoic mesenteric fat thickened and extended to the surface of the contiguous intestine, enclosing the intestinal tube. In our study, “increased” was determined as “over two-thirds of the circumference of the intestine” in transverse view.

(5) Presence of fistula signs (Figure 5). Fistula signs were defined as a hypoechoic, duct-like, peri-intestinal lesion with a diameter smaller than 2 cm, sometimes with internal echoic spots related to the presence of air, debris, or intestinal material [20].

Any discrepancies were resolved, and a final definite diagnosis was made by consultation with a senior sonographer (HY) with 10 years of experience in IBD ultrasound examination.

According to a review of our hospital’s electronic medical record system, a gastroenterologist (KY), who was blinded to the previous US examinations, created an anonymous file for each patient. The gastroenterologist collected baseline clinical information (gender, age when diagnosed as CD, duration of illness) and surgical details (indication and methods of operation). After lesion information was obtained from doctors who matched the images between TAUS and WIUS, the location of the lesions and the macroscopic changes described in the surgical report were recorded (Table 1). Then, a statistical analysis of baseline data and US features between TAUS and WIUS was performed.

### 2.6. Statistical Analysis

Taking WIUS as a standard, the sensitivity, specificity, and accuracy of TAUS were calculated according to different US parameters. The accuracy was defined as the ability to differentiate cases correctly according to the reference or standard; this was the proportion of true positives and true negatives in all evaluated cases [21]. The sensitivity, specificity, and accuracy of TAUS were calculated according to a previously published formula [21]. Descriptive statistics of baseline data included the mean and standard deviation. A two-tailed Fisher’s exact test and chi-square test were used to compare the frequencies of all categorical variables for differences between WIUS and TAUS, when appropriate. The level of significance was set at *p* < 0.05.

The agreement of image parameters between TAUS and WIUS was calculated using *kappa* statistics, which measure agreement independent of that obtained by chance. *Kappa* ranges from −1 (complete disagreement) to +1 (perfect agreement). A *kappa* value < 0 indicates no agreement, 0–0.20 indicates slight agreement, 0.21–0.40 indicates fair agreement, 0.41–0.60 indicates moderate agreement, 0.61–0.80 indicates substantial agreement, and 0.81–1.00 indicates almost perfect agreement [22]. Because it is rarely plausible that agreement might be lower than that expected by chance, we estimated the strength of agreement by constructing a confidence interval for *kappa*.

## 3. Results

### 3.1. Baseline Information

A total of 29 patients were enrolled in this study, of whom 22 were male and seven were female. The mean age at resection was 32.3 ± 10.39 (26.0–39.0) years. The median (interquartile range) of illness duration was 6.0 (2.4–12.5) years. The reason for intestinal surgery was stenosis (48.3%), refractory disease (3.4%), stenosis and fistula (27.6%), stenosis and refractory disease (17.2%), and stenosis and abscess (3.4%). The median (interquartile range) of surgery resection length was 23 (5–73) mm. Overall, 113 lesions were compared between WIUS and TAUS. The most commonly affected site was the non-terminal ileum (49.5%). The three most frequent changes in the gross morphology of specimens were fibrous stenosis (42.5%), openly longitudinal ulcerations (33.6%), and openly irregular ulcerations (20.4%). The basic characteristics of the subjects and samples enrolled in the study are shown in Table 1.

### 3.2. Evaluation of the Diagnostic Efficiency of TAUS

The consistency of images obtained from preoperative TAUS and postoperative WIUS is summarized in Table 2. Only 30 lesions were evaluated as the “existence of wall stratification” by TAUS and WIUS; therefore, only these 30 lesions were examined for the presence of submucosal thickening. The US parameter of increased fat wrapping yielded the highest sensitivity of 89.4%. The highest values of specificity (96.0%) and accuracy (92.7%) were related to the presence of fistula signs. The *kappa* coefficient for all parameters (fat wrapping, existence or disappearance of the intestinal wall, irregularity of the mucosal surface, thickening of the submucosa, and fistula signs) ranged between 0.446 and 0.615, which indicated that images of preoperative TAUS and postoperative WIUS had moderate-to-good consistency. Table 3 shows the consistency of the detailed parameters of TAUS and WIUS for the submucosal and mucosal parietal layers, with a *kappa* coefficient of 0.454.

## 4. Discussion

High sensitivity and high penetration power are the key advantages of WIUS [23]. WIUS has successfully been used to describe the US features of gastric carcinoma [24] and the intestine [14,15]. WIUS can directly scan tissues shortly after gastrointestinal tract resection, providing a clear image that avoids the possibility of errors caused by inexperienced examiners. Nevertheless, before an intestinal WIUS scan, feces and blood in the intestinal cavity should be removed, which is a difficult and time-consuming task. TAUS is commonly used for the clinical diagnosis of IBD. As doctors who work in the Inflammatory Bowel Disease Center and who have years of experience in intestinal WIUS, we wondered whether TAUS, an easier and more commonly used method than WIUS, could achieve a diagnostic efficiency similar to that of WIUS. More importantly, we investigated whether TAUS could be used to accurately examine the state of CD via complex histopathological changes. Therefore, using WIUS as a standard, we evaluated the agreement between TAUS and WIUS for several commonly used US parameters in CD.

US grayscale imaging allows for the visualization of wall stratification as a five-layer structure (Figure 6) in the normal intestine [19]. Changes in wall stratification in the submucosal and mucosal layers are not specific for the diagnosis of CD but are commonly used for CD evaluation. The earliest change caused by CD occurs in the mucosa and submucosa and consists of hyperemia and edema [25]. On the basis of many valuable positive results in the literature, the EFSUMB (European Federation of Societies for Ultrasound in Medicine and Biology) Recommendations and Clinical Guidelines for Intestinal Ultrasound in Inflammatory Bowel Diseases concluded that the loss of stratification was associated with many aspects of the disease, such as CD activity, histological inflammation, and increased surgical risk [26]. Of note, a previous study on stenosis segments of CD indicated that a loss of stratification at the segment of stenosis predicted inflammation with a low degree of fibrosis, while the presence of a stratified echogenicity pattern suggested a higher degree of fibrosis [27]. In terms of the multiple significance of intestinal wall stratification, TAUS may be an alternative tool to assist in the diagnosis and evaluation of the disease state of CD.

US has shown that, during acute inflammation, the submucosal layer is thickened by fat deposition, edema, or hemorrhage [28,29]. According to our experience and the literature [30], CD begins as inflammation in the intestinal submucosa and then crosses the intestinal wall to involve the mucosa and serosa. Even if the mucosa, as shown by endoscopy, is completely intact, potential inflammatory changes may occur in the submucosa. Therefore, thickening of the submucosal layer is sensitive for predicting potential early or subclinical lesions. A previous study reported that submucosal thickening was the initialization of chronic fibrosis. With continuous thickening of the submucosa, which might be caused by collagen deposition, the accumulation of collagen gradually leads to the loss of compliance between the submucosa and the muscularis propria [31]. Eventually, all wall layers are thickened, contributing to chronic fibrosis of the entire intestinal wall [31]. The ECCO (the European Crohn’s and Colitis Organization) Guideline/Consensus Paper in 2018 reported that bowel wall thickening, as measured by US one year after surgery, was a strong independent risk factor for early postoperative recurrence [32]. Regarding bowel wall thickening, our study yielded a high detection rate by TAUS (20/30, 66.7%) and WIUS (21/30, 70.0%). Furthermore, it had the highest *kappa* value (0.615) among all parameters, indicating that TAUS has a good ability to detect underlying acute inflammation and early fibrosis in CD.

Unlike the US parameters of intestinal wall stratification and submucosal thickening, few US-related studies have reported in detail the irregular mucosal surface pattern in CD. This sign has been generally obtained by the colonoscopic observation of a patient’s intestines in vivo [33] or by the microscopic observation of intestinal specimens in vitro [34]. The interruption or unevenness of mucosal surface continuity is thought to be mainly caused by ulcers on the mucosal surface. The British Society of Gastroenterology reporting guidelines state that an irregularity of the mucosal surface is a typical feature used to distinguish IBD from non-IBD/infective colitis [34]. However, it is not specific for a diagnosis of CD. Encouragingly, in our study, this US pattern was successfully detected by TAUS and WIUS, both with a high incidence (TAUS: 72/110, 65.5%; WIUS: 79/110, 71.8%). A moderate concordance between TAUS and WIUS (*kappa*: 0.557) and a good accuracy of TAUS (80.9%) were also obtained. Therefore, we concluded that TAUS reflects the presence of inflammation by observing irregular mucosal surfaces in the bowel wall.

In addition to detecting changes in the intestinal wall, TAUS also easily detected abnormalities beyond the intestinal wall. When CD occurs, the interface of the intestinal serosa, where the mesentery and intestinal areas merge, is blurred. Mesenteric fat is inflamed and extends beyond its normal anatomical distribution, covering the surface of the adjacent intestine. This phenomenon, termed “fat wrapping” or “creeping fat”, is a specific sign of CD [35]. The degree of fat wrapping correlates with the severity of intestinal inflammation in the contiguous intestine [35]. Chronic inflammatory stimulation produces fibrofatty proliferation of the mesenteric fat [36]. Li et al. demonstrated that when quantified using computed tomography images, mesenteric fat correlated significantly with CD disease activity, which was determined by the CD activity index and C-reactive protein [37]. Because our study shows acceptable consistency between TAUS and WIUS (0.528), the detection of “fat wrapping” by TAUS might reflect chronic inflammation of the intestinal tract.

Fistulae, which are common but important extra-intestinal complications, form as an abnormal tunnel through the intestine and into the surrounding tissue. Because of its high resolution and flexibility, endoscopic US rather than TAUS is widely used for the study of CD perianal fistulae [38]. However, CD-associated fistulae may occur in any segment of the affected bowel and exist as internal fistulae (enterocutaneous, enteroenteric, enterovaginal, and enterovesical fistulae) [39] with an incidence of 5–10% in CD patients. Endoscopic US may not be able to enter through the stenosis or obstructive segment or display the internal fistula in deep and distant intestinal segments. In our study, there were 11 cases of fistula, of which eight were in the non-terminal ileum. TAUS successfully detected most fistulae in this study (63.6%, 7/11), including those in the non-terminal ileum segment (75%, 6/8), where endoscopic US may have difficulty in scanning. Encouragingly, TAUS achieved its highest accuracy (92.7%) and moderate accordance (*kappa* = 0.596) for fistula detection compared with other US parameters. These results suggest that TAUS is useful for detecting fistulae, especially internal fistulae in CD.

Although TAUS and WIUS yielded acceptable-to-good agreement, WIUS detected more anomalies than TAUS for almost all US parameters in our study, except increased fat wrapping (Table 2). It was reported that TAUS has difficulty in detecting retroperitoneal lesions, lesions behind air-distended intestinal loops, and intestinal loops located deep in tissues [36] because the visibility is limited by sound absorption, attenuation, and artifacts in tissues such as muscle and fat. In this study, the US parameter with the worst consistency between TAUS and WIUS was the disappearance of intestinal wall stratification (*kappa* = 0.446, Table 2). Similarly, TAUS and WIUS had a relatively low consistency (*kappa* = 0.454, Table 3) for “distinct presence/indistinct presence/disappearance” of wall stratification than other US parameters, suggesting that examination of the appearance of wall stratification might easily be misdiagnosed by TAUS. A potential reason for this is that linear stratification by US is subtle and needs to be carefully identified, whereas other US parameters (submucosal thickening, irregularity of mucosal surfaces, increased fat wrapping, and fistulae) can easily be detected and determined. Therefore, we speculated that the location of lesions and the fineness of US parameters may be related to the small number of misdiagnosed cases by TAUS.

There were several limitations in this study. First, all cases enrolled were in the complex situation of refractory disease or stenosis or had fistulae, which suggest a deteriorating stage of CD. Conservatively treated cases with mild and early-stage CD might have different US appearances from those in the current study. Second, vascularity and stiffness of the intestinal wall were not evaluated in our study. It was reported that the emerging technologies of TAUS-based contrast-enhanced US and elastography US can improve the diagnostic efficiency of US by detecting the vascularity and stiffness of the intestinal wall, which helps to predict intestinal inflammatory activity and fibrosis progress [40,41]. However, because they cannot be evaluated by in vitro WIUS, it is not clear whether TAUS can accurately measure these two US parameters.

## 5. Conclusions

The results of our study demonstrate that when properly performed, the results obtained from imaging with TAUS closely approximate those with WIUS. Moreover, TAUS can be used to evaluate the relationship between US and complex intestinal conditions (acute/chronic inflammation, mild/serious fibrosis, extra-intestinal complications) in CD. Nevertheless, because it is still imperfect and has a small discrepancy compared with the actual disease characteristics obtained by WIUS, the application of TAUS should be improved further.

## Figures and Tables

**Figure 1 diagnostics-10-00267-f001:**
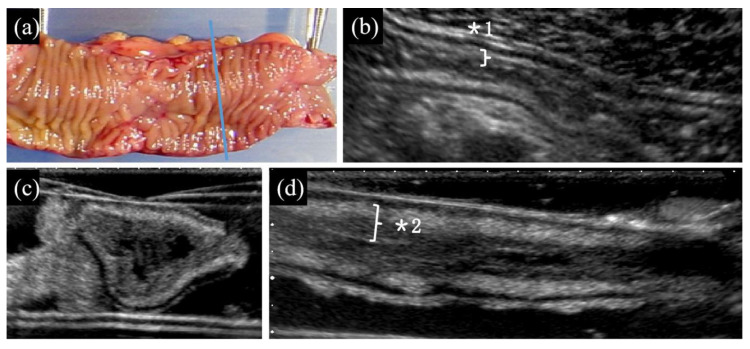
Macroscopically intact terminal ileum. (**a**) Macroscopic observation of locations in the resected bowel segment from open view (the blue line shows the scanning line of the transverse view). The thin and shallow annular folds are distributed regularly. Macroscopically, the segment is intact. No mucosal defects such as ulcers and erosion are seen. “Intact” indicates the macroscopically normal part of the surgical specimen between stenosis or a fistula, which was the reason for surgery. (**b**) Longitudinal view of TAUS. The wall stratification is clearly demonstrable in the submucosal and mucosal layers. The mucosal and submucosal layers show slight homogeneous thickening (*1). The mucous surface is regular in appearance. There is slight hypertrophy of adipose tissues around the intestinal wall, but the extent is limited (no more than half of the circumference of the intestine) and has not yet reached the degree of “fat wrapping”. Fistulae are not detected. Transverse (**c**) and longitudinal (**d**) views of the intestinal section by WIUS (closed view). The wall stratification of submucosal and mucosal layers is clearly visible. The mucosal and submucosal layers show mild homogeneous thickening (*2). The mucous surface is regular. Mild hypertrophy of the adipose tissue of the mesentery is present in the transverse view (**c**). The longitudinal view (**d**) shows that the width of the intestine is uniform, and there is liquid echogenicity in the intestinal cavity. No fistula signs are detected. For this lesion, trans-abdominal ultrasound (TAUS) and WIUS were consistent in determining the US parameters: (1) distinct presence of submucosal and mucosal layers; (2) thickening of the submucosal layer; (3) regularity of the mucous surface; (4) absence of increased fat wrapping; and (5) absence of fistula signs.

**Figure 2 diagnostics-10-00267-f002:**
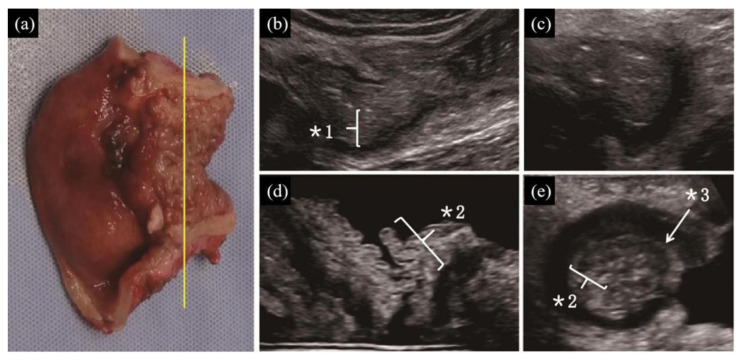
Fibrous stenosis with inflammatory polyps located in the junction of the ileocolon. (**a**) Gross morphology findings. The yellow line shows the location of the scanning area. (**b**,**c**) From longitudinal (**b**) and transverse (**c**) view by TAUS, the boundary between the mucosa and the submucosa disappear with mild increased echogenicity (*1). There is a large area of adipose tissue wrapped around the intestinal tract. (**d**,**e**) Transverse view of the affected section by WIUS (open view (**d**) and closed view (**e**)). The mucosal, submucosal, and muscular layers, as well as the entire intestinal wall, are thickened. Slight heterogeneous hyperechogenicity (mixed with some hypoechogenicity) is found in the mucosal layer, suggesting inflammatory polyps (*2). The submucosa is not very clear in the longitudinal view but is recognized in the transverse view, indicating homogeneous high echogenicity (*3). Circumferential hypertrophy of adipose tissue is seen around the intestine. For this lesion, (1) TAUS determined that the submucosal and mucosal layers had an indistinct presence, whereas WIUS identified them as distinct; (2) TAUS could not detect changes in the thickness of the submucosal layer, whereas WIUS detected thickening of the submucosal layer; (3) TAUS and WIUS were consistent in detecting irregularities of the mucosal surface; (4) TAUS and WIUS successfully detected increased fat wrapping around the bowel wall; and (5) TAUS and WIUS did not detect fistula signs.

**Figure 3 diagnostics-10-00267-f003:**
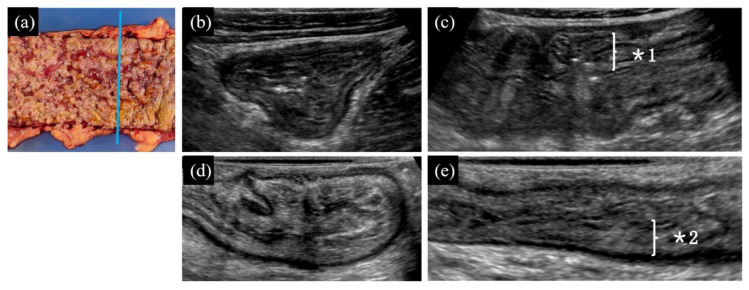
Cobblestone-like appearance in hepatic flexure of the ascending colon. Cobblestone-like appearance, which is a specific characteristic of CD, indicates longitudinal and circumferential fissures and ulcers on separate islands of mucosa, giving it an appearance reminiscent of cobblestones. (**a**) Gross morphology findings from open view. The blue line shows the scanning line of the transverse view. Note the sharp demarcation between the cobblestone mucosa of the involved segment and the grossly colonic mucosa. (**b**,**c**) From transverse (**b**) and longitudinal (**c**) views by TAUS, the appearance of the serosa and the muscular layer of the intestinal wall is observed. However, stratification of the mucosal and submucosal layers is indistinct (*1). The hyperechoic submucosal layer shows heterogeneous thickening. The mucous surface is irregular in appearance. Hyperechoic adipose tissue is present around the intestine. No fistula signs are detected. (**d**,**e**) Transverse (**d**) and longitudinal (**e**) views of the intestinal section by WIUS (closed view). The images show that the mucosal layer is blurred while the submucosal layer is heterogeneous by echogenicity with an uneven thickness (*2). The serous and muscular layers are clearly displayed with normal thicknesses. The mucous surface is irregular in appearance. There is hyperechoic adipose tissue around the intestine wall with mild hypertrophy. The continuity of the intestinal wall is good, and there is no interruption in the serous layer or the formation of hypoechoic fistulae. For this lesion, TAUS and WIUS were consistent in determining the following US parameters: (1) indistinct presence of submucosal and mucosal layers; (2) thickening of the submucosal layer; (3) irregularity of the mucous surface; (4) increased fat wrapping around the bowel wall; and (5) absence of fistula signs.

**Figure 4 diagnostics-10-00267-f004:**
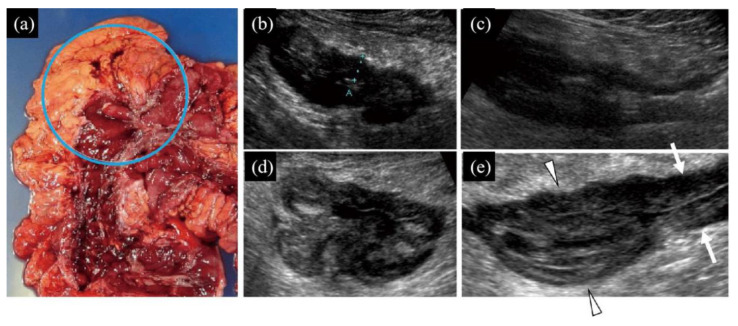
Openly irregular ulcerations and resulting stenosis located in the hepatic flexure of the colon. (**a**) Gross anatomical specimen. The blue circle shows the location of the scanning area. (**b**,**c**) Transverse (**b**) and longitudinal (**c**) views by TAUS. Bowel wall stratification in the submucosal and mucosal layers has disappeared, with extreme hypoechogenicity in the wall. The intestinal wall shows homogeneous thickness. Because the stratification of the intestinal wall has disappeared, it is impossible to determine which layer is thickened. The green dotted line on the transverse view (**b**) is the measurement of the wall thickness, which is 9 mm (normal thickness ≤ 4 mm). The mucous surface is not clearly seen, so it is difficult to judge whether it is irregular. The adipose tissue around the intestine is significantly thickened and entirely circles the intestine. (**d**,**e**) Transverse (**d**) and longitudinal (**e**) views by WIUS (closed view). The intestinal wall stratification of the submucosal and mucosal layers has disappeared. Homogeneous and unclear focal hypoechogenicity can be seen in the wall (suggestive of an ulcer). Whether the submucosal layer is thickened cannot be judged on the condition that the wall stratification of the submucosal and mucosal layers has disappeared. The mucous surface is irregular. Heterogeneous hyperechoic adipose tissue surrounding the outer wall of the intestine is highly thickened. The longitudinal (**e**) view shows that the affected intestinal segment (arrowheads) by sonography is highly thickened compared with the surrounding segment (arrows). Because of the thickened intestinal wall and narrow intestinal cavity, liquid echogenicity in the intestinal cavity did not produce clear images. The continuity of the intestinal wall is good, and there is no interruption in the serous layer or the formation of hypoechoic fistulae. For this lesion, TAUS and WIUS were consistent in determining some US parameters: (1) disappearance of wall stratification in the submucosa and mucosal layers; (2) a change in thickness of the submucosal layer could not be judged; (3) TAUS could not distinguish the mucous surface, whereas WIUS showed the mucous surface was irregular; (4) TAUS and WIUS showed agreement in detecting increased fat wrapping around the bowel wall; and (5) absence of fistula signs.

**Figure 5 diagnostics-10-00267-f005:**
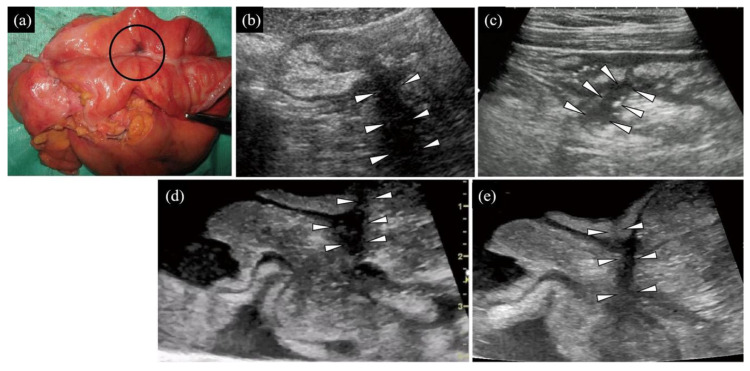
Openly longitudinal ulcer and internal fistula of the ileum. (**a**) Gross morphology findings. The black circle shows the location of the scanning area. (**b**,**c**) Transverse (**b**) and longitudinal (**c**) view by TAUS. (**d**,**e**) Transverse view by WIUS (open view). The findings of WIUS and TAUS are similar. Wall stratification of the submucosal and mucosal layers has disappeared. Because the wall stratification has disappeared, it is not possible to determine whether the submucosal layer is thickened. WIUS and TAUS detected irregularities in the mucosal surface. In the transverse view, TAUS (**b**) and WIUS (**d**,**e**) show that fat wrapping does not reach two-thirds of the circumference of the intestine. A tubular hypoechoic structure passes through and extends out of the intestinal wall (arrowhead). It is worth noting that the intestinal wall, where the tubular hypoechoic structure is located, is disorganized and indistinguishable from surrounding tissue. The intestinal canal is pulled and deformed. This abnormality is considered as an intestinal adhesion caused by chronic inflammatory changes around the fistula. For this lesion, TAUS and WIUS consistently determined the following US parameters: (1) disappearance of the submucosal and mucosal layers; (2) because the stratification disappeared, submucosal layer thickening could not be determined; (3) irregularity of the mucosal surface; (4) increased fat wrapping around the bowel wall was not detected; and (5) presence of fistula signs.

**Figure 6 diagnostics-10-00267-f006:**
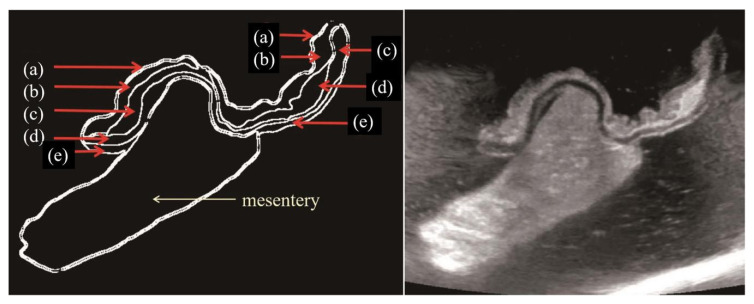
Distinct wall stratification as a five-layer structure in an ileal segment. The resected ileum was incised and opened on the anti-mesenteric border, scanned from the mucosal surface (open view) in transverse view. The right panel shows the image obtained from WIUS, and the left panel is a sketch of the stratification of the bowel wall. The first-to-fifth layers represent the border between (**a**) the lumen and mucosal layer (hyperechogenic); (**b**) the mucosal layer (hypoechogenic); (**c**) the submucosa (hyperechogenic); (**d**) the muscular layer (or the muscle membrane proper) (hypoechogenic); (**e**) and the serosa layer (hyperechogenic).

**Table 1 diagnostics-10-00267-t001:** Baseline characteristics of enrolled patients.

**Number of Patients**	**29**
Males/females	22/7
Median age, years (mean ± standard deviation) ^1^	32.3 ± 10.39
Duration of CD, years (range)	6.0 (2.4–12.5)
Mode of surgery (*n*, %)	
Partial resection of small intestine/ileocecal resection/subtotal colectomy/others ^2^	12 (41.4%)/10 (34.5%)/4 (13.8%)/3 (10.2%)
Indication of surgery (*n*, %)	
Stenosis alone/stenosis and fistula/stenosis and refractory/stenosis and abscess/refractory alone	14 (48.3%)/8 (27.6%)/5 (17.2%)/1 (3.4%)/1 (3.4%)
**Number of Lesions**	**113**
Length of excision, mm (range) ^3^	23 (5–73)
Location of resection (*n*, %)	
Non-terminal ileum/terminal ileum/colon other than cecum/cecum	56 (49.5%)/27 (23.9%)/19 (16.8%)/11 (9.7%)
Macroscopic findings of the resected lesions evaluated by WIUS ^4^ (*n*, %)	
Macroscopically intact/longitudinal ulcer scar/openly longitudinal ulcerations/fibrous stenosis/openly irregular ulcerations/cobblestone-like appearance/inflammatory polyp/internal fistula	14 (12.4%)/14 (12.4%)/38 (33.6%)/48 (42.5%)/23 (20.4%)/9 (8.0%)/12 (10.6%)/8 (7.1%)

^1^ The age here indicates the patient’s age when diagnosed with CD. ^2^ Here, “others” include ileocecal resection and partial resection of the small intestine, ileocecal resection and partial colectomy, and subtotal colectomy and partial resection of the small intestine. ^3^ Length of excision indicates the extent of involvement of the lesion. ^4^ Several lesions had more than one type of macroscopic finding. CD: Crohn’s disease. WIUS: water-immersion ultrasound.

**Table 2 diagnostics-10-00267-t002:** Agreement between TAUS and WIUS for imaging findings of CD.

Image Parameters	No. of Lesions ^1^	Both (+)	TAUS (+)	WIUS (+)	Both (−)	Sensitivity (%)	Specificity (%)	Accuracy (%)	*kappa* Values	95% CI	*p*
Disappearance (+)/presence (−) of wall stratification in submucosal and mucosal layer	113	52 (46.0%)	6 (5.3%)	25(22.1%)	30 (26.5%)	67.5	83.3	72.6	0.446	0.289–0.603	<0.0001
Thickening (+) of submucosa (calculated only when stratification is present) ^2^	30	18 (60.0%)	2 (6.7%)	3 (10.0%)	7 (23.3%)	85.7	77.8	83.3	0.615	0.311–0.919	0.001
Irregularity (+) of mucosal surface	110	65 (59.1%)	7 (6.4%)	14 (12.7%)	24 (21.8%)	82.3	77.4	80.9	0.557	0.393–0.724	<0.0001
Increase (+) in fat wrapping	104	42 (40.4%)	20 (19.2%)	5 (4.8%)	37(35.6%)	89.4	64.9	76.0	0.528	0.373–0.683	<0.0001
Presence (+) of fistulae signs	110	7 (6.4%)	4 (3.6%)	4 (3.6%)	95 (86.4%)	63.6	96.0	92.7	0.596	0.342–0.850	<0.0001

^1^ In some groups, TAUS could not determine the existence of US signs in some lesions for the possible reason of intestinal peristalsis, intestinal gas, and deeply located lesions (related to the sound attenuation of fat and muscle layers), so they were recorded as missing values and excluded from the statistical analysis. This phenomenon resulted in a frequency of observations that were less than the total number of lesions. ^2^ Thickened submucosa was calculated only when stratification was present. CI: confidence interval.

**Table 3 diagnostics-10-00267-t003:** Comparison between TAUS and WIUS for imaging findings related to detailed wall stratification in submucosal and mucosal layers (n = 113).

	Parameters	WIUS (*n*, %)			*Kappa* Values	95% CI
		Disappearance	Indistinct Presence	Distinct Presence		
**TAUS (*n*, %)**	**Disappearance**	52 (46.0%)	4 (3.5%)	2 (1.8%)	0.454	0.311–0.597
	**Indistinct Presence**	19 (16.8%)	15 (13.3%)	0 (0%)		
	**Distinct Presence**	6 (5.3%)	4 (3.5%)	11 (9.7%)		

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
