# Peer review of "Consistency of Trans-Abdominal and Water-Immersion Ultrasound Images of Diseased Intestinal Segments in Crohn’s Disease"

_diagnostics, 2020, doi:10.3390/diagnostics10050267_

Round 1
Reviewer 1 Report
This is a well-conceived study. The manuscript is finely written. The information is useful and clinically relevant.
I have only a few suggestions:
General
Since the ultrasound feature extraction was performed as follows: “Then, they identified and recorded the US changes from the images and videos of 183 TAUS and WIUS according to US parameters previously published [13, 15-17] and our clinical 184 experience in CD diagnosis..”, I think that the authors should emphasize the features that they used that differ from those that were previously published. These are new concepts and should be stated as such. This adds to the originality of the manuscript.
Specific
- Abstract -the final sentence “The images of TAUS were similar to those of WIUS, 51 suggesting that TAUS would be a suitable tool to evaluate the actual intestinal conditions of 52 CD” should be modified to say, “The images of TAUS showed substantial similarity to those of WIUS….suggesting that TAUS may function as a substitute…”
- Conclusions “we think that TAUS is similar to WIUS” needs to be changed to “the results of our study demonstrate that when properly performed, the results obtained from imaging with TAUS closely approximate WIUS”.
Author Response
We much appreciate the reviewer's positive evaluation of our work. According to your valuable comments, we made a point-to-point response in Word file in attachment. We made revision in the latest manuscript accordingly.

Reviewer 2 Report
This manuscript is an original article that retrospectively evaluated the utility of trans-abdominal ultrasound (TAUS) imaging in comparison to water-immersion ultrasound (WIUS) imaging for Crohn’s disease. The authors demonstrated the high accuracy of TAUS based on 5 US parameters, and moderate-to-good agreement between TAUS and WIUS was observed for most US parameters.
This study was conducted well, and the methods are appropriate. The data are presented clearly. In general, this is a well-written paper that presents interesting data. The results will be of interest to clinicians in the field.
However, the following major and minor issues require clarification:
Major
- The authors stated that, according to their experience, CD began as inflammation in the intestinal submucosa. The authors should reference some studies that support their hypothesis.
- The authors discussed the reason for the relatively low consistency in wall stratification, and stated that subtle US findings of linear stratification may induce a misdiagnosis. I think that the heterogeneity of the intestinal inflammation within each lesion and the difficulty in performing an objective wall assessment may also affect the low consistency.
Minor
- (P3L16) The authors should replace “[14]” with “[15]”.
- Please explain Figure 1 in Material and Methods section.
- (Table 1) The authors should replace “[minimum, maximum]” with “[range]”.
- (Table 2) The authors should modify the table as it is a little confusing in its current form.
- (P16L8) The authors should replace “TAUS” with “WIUS”.
- Please carefully proofread the manuscript before resubmission. Proofreading by a native speaker is also recommended.
I hope these comments will be helpful for improving this manuscript.
Author Response
We much appreciate the reviewer's positive evaluation of our work. According to the reviewer's insight comments, we made a detailed check and revision of the related issues. Please see attachment.

Reviewer 3 Report
In this retrospective study the authors evaluated the diagnostic accuracy of trans-abdominal ultrasound (TAUS) and water-immersion ultrasound (WIUS) in intestinal specimens of patients with Crohn's disease (CD).
Line 76-78 it should be reported that US has a role also in the evaluation of extramural complications of CD as well as a predictor of surgical recurrence after surgery (see Ribaldone et al. Scand J Gastroenterol 2013;48:552-5).
It would be interesting to understand if there are correlations between treatments used for CD abd diagnostic accuracy of the two procedures. The authors could enrich if possible, their work with this information.
The acronyms should be revised (for example US is repeated 2 times (line 79 and 87).
Some reference could be updated. For example the number 8 with another of the same authors: Sarno et al. Minerva Gastroenterol Dietol 2019;65:335-45.
Author Response
We thank the reviewer for in-depth comments and have revised the manuscript accordingly. Please see attachment. The revised portions in the manuscript have been written in red. Again, thank you very much for reviewing our manuscript.

Round 2
Reviewer 2 Report
The authors have addressed my comments and revised the manuscript enough to accept it.